# OpenReview forum: "Preemptive Detection and Steering of LLM Misalignment via Latent Reachability"
_ICLR.cc/2026/Conference — Submitted to ICLR 2026_

### Official Review · Reviewer_eDJ8 · 2025-10-30

**Soundness:** 2
**Presentation:** 2
**Contribution:** 3
**Rating:** 2
**Confidence:** 3

**Summary:**

Their method trains a classifier to determine if a partial LLM generation will be completed harmfully. In RL-BTR-ALIGN, the labels are derived using the safety of the partial generation by itself, and the discounted safety of the rest of the generation. In SAMPLE-BRT-ALIGN, the classifier is always trained to predict the safety of the final generation given the partial generation.

They also develop a method to steer the model away from the unsafe generation once it’s detected.

**Strengths:**

Their work is mostly easy to understand.

The novelty is sufficient with the reachability analysis and steering method. Overall I like the methodology of applying control theory to LLM safety.

Their inference time steering technique substantially increases safety rate, with only a small coherence cost. They also calculate results across multiple training seeds, making the significance clear.

**Weaknesses:**

I don’t think their baselines are correct in figures 2 and table 2.
Their baselines either predict everything is safe or everything is unsafe. I don’t think this is correct, because the SAP paper reports a classification accuracy of 86% for Llama2-7B on BeverTails (figure 6). (The BeaverTails dataset is relatively balanced between safe and unsafe: “44.64% were assigned the safe meta-label, while the remaining 55.36% were categorized under the unsafe meta-label.”)

Clarity

In section 4.3, they don’t explain how the argmax part of equation 1 is computed. This is a significant issue since it’s one of the main parts of the method.

The definition of ℓ(zt) is confusing. z_t is the layer-l embedding of the last emitted token, but section 5 says ℓ(zt) is calculated using a classifier given the token sequence y_0 to y_t.

They say “we assume that LLM deterministically selects the most likely token”, but it doesn't discuss the consequences of this assumption. Will their approach work with non deterministic decoding strategies?

Minor points

The paper could better motivate why it’s necessary to classify partial generations as leading to unsafe outputs. For example, an alternative would be to generate the entire response from the LLM, and then classify the entire response as safe or unsafe before showing it to the user, or acting on it. The paper could mention that needing to wait until the entire response is generated before classifying will increase streaming latency.

**Questions:**

none

---

> ### Author Response · Authors · 2025-11-20
> **Author Rebuttal [Part 1]**
>
> We thank the reviewer for the feedback. We appreciate the words about our novel methodology and experimental findings showing that BRT-Align is capable of substantially improving the safety rate at a minor coherence cost and at a much faster rate.
>
> > I don’t think their baselines are correct in figures 2 and table 2. Their baselines either predict everything is safe or everything is unsafe. I don’t think this is correct, because the SAP paper reports a classification accuracy of 86% for Llama2-7B on BeverTails (figure 6). (The BeaverTails dataset is relatively balanced between safe and unsafe: “44.64% were assigned the safe meta-label, while the remaining 55.36% were categorized under the unsafe meta-label.”)
>
> We appreciate the opportunity to clarify why our findings in Figure 2/Table 2 seem to differ from Figure 6 of the Safety Polytope (SaP) paper. There are two main factors that explain the performance differences: (1) a bug uncovered in our F1 score computation for runtime monitoring, and (2) fundamental differences in what is being evaluated at test-time. We address both below and explain why these differences fully account for the performance differences.
>
> **(1) Bug in our F1 Score Computation.** Following the reviewer’s comment, we re-audited our evaluation pipeline and found a bug in the F1 score computation. Specifically, we had inadvertently set the precision as the true positive rate and the recall as the true negative rate. This bug resulted in a considerably lower F1 score for SaP, while only slightly affecting the Re-Control baseline and our methods. For transparency, we have updated the supplementary code by updating the affected file `/runtime_monitoring/plot_balanced_runtime_monitor_results.py`, with an associated comment on Line 48.
>
> After fixing the bug, SaP’s average F1 score is ~0.67 across LLMs (significantly higher than initial reported numbers). This value corresponds to a true positive rate (predicting unsafe) of 100% and a true negative rate of ~1% (predicting safe).  Importantly, the relative ranking of methods and conclusions of our work remain unchanged – BRT-Align (our) methods still consistently outperform (or perform equal to) the baselines. We have also updated all metrics in the revised manuscript accordingly.
>
> **(2) Test-time Evaluation Differences (explains why our F1 scores are lower than those reported in the SaP paper, even after fixing the bug).** The reviewer’s concern is motivated by the 86% accuracy reported in the SaP paper. However, this number reflects a different evaluation setup and a different prediction target between our paper and the SaP paper, leading to a mismatch between the reported numbers.
>
> First, the reviewer notes that SaP reports 86% accuracy on BeaverTails. However, our usage of the BeaverTails dataset differs from theirs in a few important aspects.
>
> (2a) The SaP paper evaluates on BeaverTails’ ground-truth responses; we evaluate on each LLM’s own generated responses.
> - **SaP Paper** utilizes the BeaverTails test dataset’s *provided* responses and labels for evaluation. Moreover, every LLM is evaluated on the same fixed outputs, independent of what the model would actually generate.
> - **Our Paper** also uses BeaverTails test dataset prompts, but each LLM generates its *own responses*. These responses are labeled using a RoBERTa toxicity classifier. All methods (including the SaP baseline and our methods) are evaluated against these labels to measure accuracy, precision, and other relevant metrics. The reason for this experimental choice is that we are ultimately interested in monitoring the *actual generation* of a given LLM, rather than the response provided in the BeaverTails dataset.
>
> Thus, the 86% accuracy in SaP reflects performance on BeaverTails’ curated responses, not on *model-generated* outputs. Because our outputs differ (often significantly) from BeaverTails responses, the reported accuracy/F1 in the SaP paper cannot be directly compared to that in our paper—hence the mismatch the reviewer observed.
>
> (2b) SaP performs post-hoc whole-output classification; we evaluate *first-token unsafety detection.*
>
> - **SaP Paper** evaluates over the test set of prompts and responses from the BeaverTails dataset (from Page 7 of their paper: “We measure classification accuracy on BeaverTails’ test set, reporting the mean and standard deviation of per category accuracy across categories, repeated over 5 seeds.”). This evaluation is performed on *full responses*, using a single safe/unsafe label per entire output (i.e., post-hoc classification).
> - **Our Paper:** In contrast, our evaluation measures when a token-level safety monitor *first* detects unsafety during generation, which is a much stricter criterion. This is because our evaluation stresses early detection and steering of misalignment, not post-hoc correctness.
>
> [This response continues to the next part.]

---

> ### Author Response · Authors · 2025-11-20
> **Author Rebuttal [Part 2]**
>
> > I don’t think their baselines are correct in figures 2 and table 2. Their baselines either predict everything is safe or everything is unsafe. I don’t think this is correct, because the SAP paper reports a classification accuracy of 86% for Llama2-7B on BeverTails (figure 6). (The BeaverTails dataset is relatively balanced between safe and unsafe: “44.64% were assigned the safe meta-label, while the remaining 55.36% were categorized under the unsafe meta-label.”)
>
> [Continued Response from Part 1]
>
> To highlight the effect of this difference: If we evaluate SaP only on the final token (i.e., mimic their setting), SaP achieves a F1 score of 0.9647 ± 0.0155 across 5 seeds for Llama2-7B – higher than the ~0.67 reported in Figure 2 and closer to their 86–90% reported accuracy.
> The lower F1 (approximately 0.67) in Figure 2 reflects our token-level early-detection setup, not final-response classification. This explains why SaP performs poorly even though its final-token accuracy is high.
>
> Thus, the direction of the discrepancy is fully explained by the difference in evaluation goals: **the SaP paper measures post-hoc correctness on BeaverTails’ ground-truth responses, while we measure early-warning performance on actual LLM responses.** To further validate our understanding, we also replicated SaP’s BeaverTails evaluation using their setup. For Llama2-7B, we obtained a classification accuracy of 72.7% (seed=42), which falls within their reported error bars.
>
> **Summary.** In summary, the low F1 score for the SaP baseline in our original submission is due to (i) a metrics bug that understated SaP’s performance, and (ii) recognizing that SaP’s reported 86% accuracy and our reported F1 scores are not directly comparable because the underlying evaluation tasks differ in both dataset construction and detection objective. We have updated all affected metrics and will clarify these differences explicitly in the revised manuscript (Section “Dataset Construction” and “Evaluation Protocols”).
>
> > In section 4.3, they don’t explain how the argmax part of equation 1 is computed. This is a significant issue since it’s one of the main parts of the method.
>
> We appreciate the opportunity to clarify the argmax in Equation 1. In Section 4.3, we state that “$\epsilon$ is sampled from an $L^2$-norm ball $B_R(\mathbf{0})$ with radius $R$ and center $\mathbf{0}$.” Specifically, at the latent state $z$, we sample N-perturbations, $\epsilon_1, \epsilon_2, \ldots, \epsilon_N$, uniformly randomly from $B_R(\mathbf{0})$. Correspondingly, values of the perturbed latent states are given by $V(z + \epsilon_1), V(z + \epsilon_2), \ldots, V(z + \epsilon_N)$. Finally, the argmax is computed as $\epsilon^* = \arg \max_{i=1:N} V(z + \epsilon_i)$. We have now added a sentence in the revised manuscript to clarify that “We compute the $\arg\max$ by evaluating the safety value function with the perturbed latent states and select the $\epsilon$ with the highest value.”
>
> In our work, we sample 1000 $\epsilon$ values in the $L^2$-norm ball to perform the maximization (i.e., $N=1000$). We have updated Appendix 11.7 to make this clear in the manuscript. To contextualize the sampling efficiency of searching for latent embeddings with this scheme, for Qwen2-1.5B, the total average token inference time is ~0.015 seconds per batch of 32 prompts, of which the time spent in sampling latent perturbations and computing the argmax is ~0.0015 seconds. In other words, the time needed for sampling latent embeddings with higher value is insignificant compared to the overall time spent for LLM token generation.

---

> ### Author Response · Authors · 2025-11-20
> **Author Rebuttal [Part 3]**
>
> > The definition of ℓ(zt) is confusing. $z_t$ is the layer-l embedding of the last emitted token, but section 5 says ℓ(zt) is calculated using a classifier given the token sequence $y_0$ to $y_t$.
>
> We appreciate the reviewer’s question and agree that the original notation is ambiguous. The key point is that although $z_t$​ denotes the layer-$l$ embedding of the most recently generated token, the score $\ell(z_t)$ is not computed from this vector in isolation. Instead, it is computed from the *entire token prefix* $(y_0,\dots,y_t)$ that produced $z_t$. The classifier processes this prefix using the LLM’s hidden states, and then outputs a scalar safety score. We denote this score as $\ell(z_t)$ because the score is aligned with—and used to monitor—the embedding at step $t$. In other words:
>
> - $z_t$​ = LLM’s layer-$l$ hidden state for the last emitted token $y_t$​
> - Classifier input = full prefix $(y_0, \dots, y_t)$
> - $\ell(z_t)$ = classifier output associated with the hidden state at step $t$
>
> Thus, $\ell(\cdot)$ is a function applied at time $t$ but computed using the hidden states derived from the prefix, not from $z_t$​ in isolation. Our notation overloaded $\ell(z_t)$ to indicate that the safety score is tied to the new token at time $t$, but we agree this can be misinterpreted as taking $z_t$​ alone as input.
> We have revised the manuscript to clarify this distinction and now explicitly state that while the score is aligned with $z_t$​, it is computed from the full prefix. We appreciate the reviewer’s feedback.
>
> > They say “we assume that LLM deterministically selects the most likely token”, but it doesn't discuss the consequences of this assumption. Will their approach work with non deterministic decoding strategies?
>
> We appreciate the reviewer’s thoughtful point regarding the deterministic (greedy)-decoding assumption. Our theoretical analysis indeed models the LLM’s latent dynamics under a fixed decoding rule, $z_{t+1} = f(z_t)$, which makes the backward-reachable tube tractable. This is consistent with prior work in control-theoretic LLM alignment and with the baselines we compare against, all of which are also evaluated under greedy decoding. Thus, our empirical comparisons isolate the effect of the reachability formulation rather than differences in decoding policy.
>
> That said, the algorithm itself does not rely on deterministic decoding. The learned safety value function $V(z_t)$ depends only on the latent state at timestep $t$, and can therefore be applied unchanged even when tokens are sampled using top-k, nucleus sampling, or any other stochastic decoding rule. In a stochastic setting, the latent dynamics naturally become $z_{t+1} = f(z_t, \omega_t)$, where $\omega_t$​ models sampling randomness; in principle, this leads to robust or stochastic-reachability extensions.
>
> Following the reviewer’s suggestion, we ran an additional experiment to evaluate BRT-Align under stochastic decoding. Using top-p = 0.95 nucleus sampling and temperature = 0.7 with Qwen2-1.5B, we constructed a new dataset from BeaverTails consisting of 5 sampled responses per prompt (with RoBERTa labels) and trained Sample-BRT-Align on this stochastic dataset. In this case, the value function predict the expected safety value of the LLM generation (as opposed to the worst-case safety value of the LLM generation under greedy decoding). We then evaluated the learned value function on all prompts that originally produced unsafe outputs under greedy decoding, again using top-p sampling at test-time. Averaged over 3 datasets and 5 seeds, we obtain:
>
> | Average Safety Rate | Average Cosine Similarity | Average Word Diversity |
> |---------------------|---------------------------|------------------------|
> | 0.924 ± 0.010       | 0.399 ± 0.006             | 0.900 ± 0.006          |
>
> These results indicate that BRT-Align naturally extends to stochastic decoding, achieving substantial safety improvements with modest degradation in coherence.
>
> Finally, while our current algorithm handles stochastic decoding effectively by estimating the expected safety value under the sampling distribution, more conservative formulations may ultimately be preferable. Prior work in safety-critical control (e.g., ISAACS [1]) constructs worst-case value functions over disturbance distributions to guarantee safety under the most adverse realizations. Extending RL-BRT-Align to learn such robust value functions is a natural next step—particularly for providing stronger safety guarantees under high-entropy decoding policies.
>
> We have added a subsection to the Appendix describing the stochastic-decoding experiment and clarified the scope and limitations of our theoretical assumptions in the revised manuscript.
>
> **References:**
>
> [1] Kai-Chieh Hsu, Duy Phuong Nguyen, and Jaime Fernandez Fisac. Isaacs: Iterative soft adversarial actor-critic for safety. In Learning for Dynamics and Control Conference, pp. 90–103. PMLR, 2023.

---

> ### Author Response · Authors · 2025-11-20
> **Author Rebuttal [Part 4]**
>
> > The paper could better motivate why it’s necessary to classify partial generations as leading to unsafe outputs. For example, an alternative would be to generate the entire response from the LLM, and then classify the entire response as safe or unsafe before showing it to the user, or acting on it. The paper could mention that needing to wait until the entire response is generated before classifying will increase streaming latency.
>
> Thank you for the helpful suggestion. We agree that the manuscript should better motivate why classifying only the final response is insufficient. We have updated Section 4.2 to clarify that waiting until the full output is generated introduces two key problems: (i) increased streaming latency (as the reviewer correctly notes), and (ii) the possibility that unsafe content is produced before the classifier intervenes, even if the final response is ultimately rejected.
>
> Beyond these concerns, full-response classification also provides no immediate mechanism for recovery—once the model has generated an unsafe continuation, the system must discard it and restart, rather than steering the model back toward a safe trajectory. In contrast, our method is explicitly designed to detect unsafe behavior during generation and redirect the model before harmful content is emitted.
>
> To reflect these points earlier, we have also added the following sentence to the Introduction:
>
> “Our goal is to enable *runtime safety*: only classifying completed responses, we detect and steer unsafe trajectories while the response is still being generated”
>
> We believe these revisions more clearly explain why partial-generation monitoring is necessary and why full-response post-hoc classification is not sufficient for real-time safe deployment.

---

### Official Review · Reviewer_yGuh · 2025-10-31

**Soundness:** 2
**Presentation:** 3
**Contribution:** 3
**Rating:** 6
**Confidence:** 2

**Summary:**

This paper proposes BRT-ALIGN, a reachability-based framework to conduct safety alignment for LLM during inference time. It models the autoregressive generation of LLM as a dynamical system in latent space and then learn a safety value function via backward reachability. Building on this, BRT-ALIGN is able to forecast the unsafe tokens during inference on-the-fly and take proactive intervention. It can also conduct a guided-perturbation on the latent states to steer the generation away from unsafe regions. Evaluation results on 5 LLMs demonstrates the effectiveness of BRT-ALIGN on improving safety alignment compared to baseline approaches.

**Strengths:**

- The paper studies the safety alignment of LLMs, which is a critical problem for the community.
- The proposed BRT-ALIGN brings the control theory and inference-time safety alignment, which is novel and very interesting.
- The evaluation results demonstrate the effectiveness of BRT-ALIGN, compared to other control-based alignment approaches.

**Weaknesses:**

- Lack of comparison on several SOTA alignment techniques.
- Lack of comparison on jailbreak attacks.
- The preservation of utility after applying BRT-ALIGN shall be justified by evaluating on more challenging benchmark.

**Questions:**

I think it is an interesting paper that introduces a novel technique. However, my main concerns lie in the evaluation section, which could be expanded to better contextualize the performance relative to existing methods.

1. Similar properties, such as preemptive detection, have been discussed in prior none control-based alignment approaches, for example, Circuit Breaker [1]. A direct comparison would help clarify the unique contribution of this work.

2. Currently, the evaluation is primarily conducted on raw harmful prompts. It would strengthen the paper to also test the proposed method against harmful prompts generated by jailbreak techniques such as GCG [2], PAIR [3], and AutoDAN [4].

3. To more comprehensively assess the utility preservation of BRT-ALIGN, I recommend including results from benchmarks designed to measure over-refusal behaviors, such as ORBench [5] and XSTest [6].

---
Reference
---

[1] Zou A, Phan L, Wang J, et al. Improving alignment and robustness with circuit breakers[J]. Advances in Neural Information Processing Systems, 2024, 37: 83345-83373.

[2] Zou A, Wang Z, Carlini N, et al. Universal and transferable adversarial attacks on aligned language models[J]. arXiv preprint arXiv:2307.15043, 2023.

[3] Chao P, Robey A, Dobriban E, et al. Jailbreaking black box large language models in twenty queries[C]//2025 IEEE Conference on Secure and Trustworthy Machine Learning (SaTML). IEEE, 2025: 23-42.

[4] Liu X, Xu N, Chen M, et al. Autodan: Generating stealthy jailbreak prompts on aligned large language models[J]. arXiv preprint arXiv:2310.04451, 2023.

[5] Röttger P, Kirk H R, Vidgen B, et al. Xstest: A test suite for identifying exaggerated safety behaviours in large language models[J]. arXiv preprint arXiv:2308.01263, 2023.

[6] Cui J, Chiang W L, Stoica I, et al. Or-bench: An over-refusal benchmark for large language models[J]. arXiv preprint arXiv:2405.20947, 2024.

---

> ### Author Response · Authors · 2025-11-20
> **Author Rebuttal [Part 1]**
>
> > Similar properties, such as preemptive detection, have been discussed in prior none control-based alignment approaches, for example, Circuit Breaker [1]. A direct comparison would help clarify the unique contribution of this work.
>
> We thank the reviewer for pointing us to Circuit Breakers (Zou et al., 2024). While related in spirit, Circuit Breakers targets a different problem and deployment setting than our work. Circuit Breakers is a training-time defense: it trains LoRA adapters on model-specific harmful/retain datasets to obtain a new, modified model with lower attack success. In contrast, our goal is to monitor a fixed model by analyzing its latent dynamics and predicting when unsafe generations become reachable, which is especially relevant when the base model cannot be retrained (e.g., proprietary or centrally managed models). A direct comparison would therefore mix two orthogonal axes — how to best retrain a safer model versus how to provide runtime monitoring for a given model — and is outside the scope of this work. We have made this distinction clearer in the “Related Works” section and note that our reachability-based monitor is complementary to Circuit Breakers and could, in principle, also be applied on top of a Circuit Breaker-trained model.
>
> > Currently, the evaluation is primarily conducted on raw harmful prompts. It would strengthen the paper to also test the proposed method against harmful prompts generated by jailbreak techniques such as GCG [2], PAIR [3], and AutoDAN [4].
>
> We thank the reviewer for this suggestion to strengthen our evaluation. We would like to clarify that our evaluation already includes jailbreak-generated harmful prompts. Specifically, the UltraSafety dataset used in our evaluation already includes 830 prompts (out of 3000 prompts) generated using the AutoDAN jailbreak technique (please see data documentation for more details: https://huggingface.co/datasets/openbmb/UltraSafety#ultrasafety).
>
> For the UltraSafety dataset and the base Qwen2-1.5B LLM, we evaluated the new safety rate on the prompts that originally led to unsafe responses (i.e., a 0% safety rate for the base LLM).
> We evaluated all baselines on three metrics: average safety rate, cosine similarity (coherence), and response diversity. Each metric is out of 1.  We find that Sample-BRT-Align and RL-BRT-Align lead to significantly higher average safety rates than the baselines, while preserving coherence and increasing diversity.
>
> | Method           | Average Safety Rate | Average Cosine Similarity | Average Response Diversity |
> |------------------|---------------------|---------------------------|----------------------------|
> | SaP              | 0.067 ± 0.033       | 0.591 ± 0.004             | 0.308 ± 0.012              |
> | Re-Control       | 0.368 ± 0.222       | 0.607 ± 0.011             | 0.322 ± 0.020              |
> | Sample-BRT-Align | 0.924 ± 0.031       | 0.558 ± 0.015             | 0.446 ± 0.013              |
> | RL-BRT-Align     | 0.908 ± 0.021       | 0.524 ± 0.028             | 0.440 ± 0.029              |
>
> These results provide evidence that our method acts as a strong defense against AutoDAN-style jailbreaking.
> In response to the reviewer’s comment, we have included UltraSafety dataset-evaluation tables for all models (Qwen2-1.5B, Llama2-7B, Ministral-8B, Falcon-7B, gpt-oss-20B) in the Appendix. Additionally, we have clarified in Appendix §11.1 (referenced in Section 5: “Models and Datasets”) that UltraSafety contains AutoDAN-generated prompts.
>
> We thank the reviewer again for this suggestion; we believe the clarified discussion and expanded tables make the evaluation more transparent and comprehensive.

---

> ### Author Response · Authors · 2025-11-20
> **Author Rebuttal [Part 2]**
>
> > To more comprehensively assess the utility preservation of BRT-ALIGN, I recommend including results from benchmarks designed to measure over-refusal behaviors, such as ORBench [5] and XSTest [6].
>
> We thank the reviewer for raising the important point of evaluating utility preservation and potential over-refusal. We fully agree that assessing whether a safety method leads to unnecessary refusals is crucial for understanding its practical viability.
>
> Following the reviewer’s suggestion, we evaluated over-refusal using the XSTest benchmark. We tested the gpt-oss-20b model with our most conservative variant (RL-BRT-Align) and manually annotated the model outputs for all 450 XSTest prompts. Each response was labeled as “comply,” “refuse to comply,” or “neither” (e.g., answering an unrelated question or producing incoherent text). The results are as follows:
>
> - **Base LLM:** 308 comply, 130 refuse, 12 neither
> - **RL-BRT-Align:** 305 comply, 135 refuse, 10 neither
>
> Breaking this down by safe vs. unsafe prompts:
>
> **Safe prompts:**
> - Base model compliance: 97.6%
> - RL-BRT-Align compliance: 97.2%
>
> **Unsafe prompts:**
> - Base model compliance: 32%
> - RL-BRT-Align compliance: 31%
>
> Overall, the compliance and refusal rates remain nearly unchanged under RL-BRT-Align. This indicates that our method does not introduce meaningful over-refusal on XSTest, even under the most conservative steering policy.
> We appreciate the reviewer’s suggestion and have added a discussion of these findings to the Appendix, noting that extending evaluation to other utility-preservation benchmarks such as ORBench is a valuable direction for future work.

---

### Official Review · Reviewer_c3nM · 2025-10-31

**Soundness:** 3
**Presentation:** 3
**Contribution:** 3
**Rating:** 6
**Confidence:** 3

**Summary:**

The paper centres on introducing a reachability-based framework, inspired by control theory, to mitigate unsafe LLM responses by predicting potential harmful continuations and steering the model toward safer outputs, a method referred to as BRT-ALIGN. It positions this as the first application of reachability analysis for inference-time LLM safety, offering a principled approach to both anticipate and prevent harmful text generation.

**Strengths:**

- The paper is well-structured and clearly written, with helpful figures and explanations that make the novel reachability-based approach easy to follow.

- The approach is quite novel as this paper claims to be the first to apply backward reachability in LLM latent space, enabling formal prediction and prevention of unsafe continuations.

- The paper presents a comprehensive safety framework that integrates a reachability-based monitor for early detection with a lightweight steering filter that subtly guides the model’s latent states away from unsafe regions.

- The inclusion of multiple evaluation metric provides a holistic picture of how the proposed method outperforms the existing strategies

**Weaknesses:**

- There exist some methodologies that have worked on safe generation of LLMs at inference time (Safe Infer : https://arxiv.org/abs/2406.12274, Safe Decoding : https://arxiv.org/abs/2402.08983, Self-CD : https://arxiv.org/abs/2401.17633) Including these as baselines would strengthen the comparison and contextualize the proposed approach more comprehensively.

- One of the primary assumptions of the framework is greedy decoding (line 144) to create a deterministic latent trajectory. This removes randomness and makes reachability tractable, but it raises concerns as many LLM applications use sampling (top-k, nucleus) for diversity. It is unclear how well BRT-Align would generalize when the model samples instead of greedily choosing tokens. This assumption may limit real-world applicability.

- The evaluation focuses solely on offensive/toxic language as the failure mode. The failure set is defined via an offensive-language classifier. While toxicity is important, LLM misalignment encompasses other harms (misinformation, self-harm, illegal advice, privacy breaches, etc.). The paper acknowledges this limitation, but currently BRT-Align is only validated on one dimension of harm. Its effectiveness for other harmful content is untested.

- It would be interesting to see if this methodology can somehow be modified to include proprietary LLMs as currently it is limited to open source models.

- The evaluation uses automatic metrics (classifier labels for safety, cosine similarity for coherence, n-gram diversity). While these are reasonable, human evaluation of output quality and safety would strengthen the validation.

- In some places, few notations are not previously stated making it a bit difficult to follow. For instance., W in line 134, fLM  in line 135, etc

**Questions:**

Please refer to the weakness section and address those concerns

---

> ### Author Response · Authors · 2025-11-20
> **Author Rebuttal [Part 1]**
>
> > There exist some methodologies that have worked on safe generation of LLMs at inference time (Safe Infer : https://arxiv.org/abs/2406.12274, Safe Decoding : https://arxiv.org/abs/2402.08983, Self-CD : https://arxiv.org/abs/2401.17633) Including these as baselines would strengthen the comparison and contextualize the proposed approach more comprehensively.
>
> We thank the reviewer for pointing out these relevant inference-time safety methods. SafeDecoding (Xu et al., 2024), Self-Contrastive Decoding (Self-CD; Shi et al., 2024), and SafeInfer (Banerjee et al., 2025) all improve safety by *modifying the next-token distribution during decoding.* For example, SafeDecoding and SafeInfer amplify the probability of safety disclaimers and down-weight tokens aligned with harmful continuations, while Self-CD contrasts outputs from different safety prompts to reduce over-refusal on benign inputs. In contrast, our work treats the LLM under a fixed decoding rule (e.g., greedy decoding) as a deterministic dynamical system in the latent embedding space and learns a value function that approximates the backwards reachable tube of future unsafe states. In this sense, our method provides *worst-case safety guarantees* in the latent space over future generations of LLM for a given decoding scheme, rather than introducing a new decoding heuristic. Because of this difference in focus, these inference-time decoding methods are largely complementary rather than direct competitors: our reachability-based monitor could, in fact, be applied on top of SafeDecoding, Self-CD, or SafeInfer to further enhance safety.
>
> Additionally, SafeDecoding and SafeInfer both rely on auxiliary safety-specialized models, and Self-CD is evaluated primarily on over-refusal (over-safety) benchmarks, whereas our experiments deliberately compare against baselines that (1) use the same supervision as our learned value function and (2) operate under the same deterministic decoding regime. This ensures that the performance differences we report can be attributed to the reachability formulation itself. Incorporating SafeDecoding / Self-CD / SafeInfer as full baselines in our setting would require re-implementing their additional pipelines (including extra expert/harmful models and datasets) and adapting them to our tasks, which is unfortunately beyond the scope of what we can reasonably add during the rebuttal period.
>
> As per the reviewer’s suggestion, we have updated the Related Work section in the revised draft to explicitly discuss these decoding-based safety methods and clarify how our approach relates to—and can complement—them by offering an additional layer of preemptive detection and steering. We also identify additional experimentation with such combinations as an exciting direction for follow-up work.

---

> ### Author Response · Authors · 2025-11-20
> **Author Rebuttal [Part 2]**
>
> > One of the primary assumptions of the framework is greedy decoding (line 144) to create a deterministic latent trajectory. This removes randomness and makes reachability tractable, but it raises concerns as many LLM applications use sampling (top-k, nucleus) for diversity. It is unclear how well BRT-Align would generalize when the model samples instead of greedily choosing tokens. This assumption may limit real-world applicability.
>
> We appreciate the reviewer’s thoughtful point regarding the greedy-decoding assumption. Our theoretical analysis indeed models the LLM’s latent dynamics under a fixed decoding rule, $z_{t+1} = f(z_t)$, which makes the backward-reachable tube tractable. This is consistent with prior work in control-theoretic LLM alignment and with the baselines we compare against, all of which are also evaluated under greedy decoding. Thus, our empirical comparisons isolate the effect of the reachability formulation rather than differences in decoding policy.
>
> That said, the algorithm itself does not rely on greedy decoding. The learned safety value function $V(z_t)$ depends only on the latent state at timestep $t$, and can therefore be applied unchanged even when tokens are sampled using top-k, nucleus sampling, or any other stochastic decoding rule. In a stochastic setting, the latent dynamics naturally become $z_{t+1} = f(z_t, \omega_t)$, where $\omega_t$​ models sampling randomness; in principle, this leads to robust or stochastic-reachability extensions.
>
> Following the reviewer’s suggestion, we ran an additional experiment to evaluate BRT-Align under stochastic decoding. Using top-p = 0.95 nucleus sampling and temperature = 0.7 with Qwen2-1.5B, we constructed a new dataset from BeaverTails consisting of 5 sampled responses per prompt (with RoBERTa-generated labels) and trained Sample-BRT-Align on this stochastic dataset. In this case, the value function predicts the expected safety value of the LLM generation (as opposed to the worst-case safety value of the LLM generation under greedy decoding). We then evaluated the learned value function on all prompts that originally produced unsafe outputs under greedy decoding, again using top-p sampling at test-time. Averaged over 3 datasets and 5 seeds, we obtain:
>
> | Average Safety Rate | Average Cosine Similarity | Average Word Diversity |
> |---------------------|---------------------------|------------------------|
> | 0.924 ± 0.010       | 0.399 ± 0.006             | 0.900 ± 0.006          |
>
> These results indicate that BRT-Align naturally extends to stochastic decoding, achieving substantial safety improvements with modest degradation in coherence.
>
> Finally, while our current algorithm handles stochastic decoding effectively by estimating the expected safety value under the sampling distribution, more conservative formulations may ultimately be preferable. Prior work in safety-critical control (e.g., ISAACS [1]) constructs worst-case value functions over disturbance distributions to guarantee safety under the most adverse realizations. Extending RL-BRT-Align to learn such robust value functions is a natural next step—particularly for providing stronger safety guarantees under high-entropy decoding policies.
>
> We have added a subsection to the Appendix describing the stochastic-decoding experiment and clarified the scope and limitations of our theoretical assumptions in the revised manuscript.
>
> **References:**
>
> [1] Kai-Chieh Hsu, Duy Phuong Nguyen, and Jaime Fernandez Fisac. Isaacs: Iterative soft adversarial actor-critic for safety. In Learning for Dynamics and Control Conference, pp. 90–103. PMLR, 2023.

---

> ### Author Response · Authors · 2025-11-20
> **Author Rebuttal [Part 3]**
>
> > The evaluation focuses solely on offensive/toxic language as the failure mode. The failure set is defined via an offensive-language classifier. While toxicity is important, LLM misalignment encompasses other harms (misinformation, self-harm, illegal advice, privacy breaches, etc.). The paper acknowledges this limitation, but currently BRT-Align is only validated on one dimension of harm. Its effectiveness for other harmful content is untested.
>
> We agree with the reviewer that offensive or toxic language represents only one dimension of LLM harm, and we explicitly acknowledge this limitation in the manuscript. Importantly, however, the proposed framework itself is not tied to toxicity. The reachability formulation and the safety value function $V(z_t)$ depend only on having a binary (or continuous) harmfulness signal, e.g., a classifier $\ell(z_t)$ that identifies harmful states. Mathematically, the backward reachable tube and all associated analysis remain unchanged when $\ell(\cdot)$ is replaced with a classifier for any other harm category, including misinformation, self-harm content, illegal advice, privacy leaks, etc. Extending BRT-Align to these settings would require training a new safety classifier and repeating the same workflow for constructing the value function.
>
> Nevertheless, we completely agree with the reviewer and view extending BRT-Align to multiple harm categories as an important direction for future work.
>
> > It would be interesting to see if this methodology can somehow be modified to include proprietary LLMs as currently it is limited to open source models.
>
> We appreciate the reviewer’s point on including evaluation on proprietary LLMs. As we mentioned in the Limitations section, our work largely focuses on open-source models, as we require the latent LLM encodings for training and evaluating our method. For future work, we welcome collaborations with LLM developers to obtain access to proprietary LLMs and evaluate BRT-Align in real-world settings. Beyond the issue of model access, there is nothing fundamentally limiting BRT-Align from evaluation with proprietary LLMs.
>
> > The evaluation uses automatic metrics (classifier labels for safety, cosine similarity for coherence, n-gram diversity). While these are reasonable, human evaluation of output quality and safety would strengthen the validation.
>
> We agree with the reviewer that human evaluation would provide valuable complementary evidence on both safety and generation quality. However, conducting a rigorous human study—including annotator training, multi-rater protocols, and safety review—was, unfortunately, not feasible within the rebuttal timeline. Following common practice in alignment and safety research, we therefore rely on automatic metrics that are (i) standard in the literature, (ii) reproducible, (iii) can be used to directly compare the proposed approach with existing methods; and (iv) directly aligned with the risks our method targets. In particular, classifier-based safety labels are widely used in toxicity and safety benchmarks, cosine similarity is a standard measure of semantic coherence, and n-gram diversity is commonly used to assess lexical variation.
>
> We have clarified this rationale in the revised manuscript and added to the Limitations section that a structured human evaluation—covering both safety judgments and perceived output quality—is an important avenue for future work and would further strengthen our empirical validation.
>
> > In some places, few notations are not previously stated making it a bit difficult to follow. For instance., W in line 134, fLM in line 135, etc
>
> We thank the reviewer for highlighting the missing notation. We have now added explicit definitions for $W$ and $f_{LM}$​, which were previously undefined. We also reviewed the remainder of the notation throughout the manuscript and verified that all other symbols are introduced before use. We appreciate the reviewer’s attention to clarity and have updated the text accordingly.

---

### Official Review · Reviewer_caSc · 2025-11-03

**Soundness:** 3
**Presentation:** 3
**Contribution:** 2
**Rating:** 6
**Confidence:** 4

**Summary:**

This paper introduces BRT-ALIGN, a novel framework for inference-time safety alignment of LLMs. The core idea is to model LLM token generation as a dynamical system in a latent embedding space and apply reachability analysis, a technique from control theory. By computing a "backward reachable tube" (BRT), the method can identify latent states that will inevitably lead to unsafe completions (e.g., toxic or harmful content). This enables a steering mechanism that minimally perturbs the latent state to guide generation toward safe outputs. The authors propose two variants for learning the required safety value function: RL-BRT-ALIGN (using a Bellman recursion) and SAMPLE-BRT-ALIGN (a simpler supervised approach). Through comprehensive experiments on five open-source LLMs and three safety benchmarks, the paper demonstrates that BRT-ALIGN significantly outperforms existing control-theoretic baselines in both the accuracy of unsafe content detection and the effectiveness of safety steering, all while incurring lower computational overhead.

**Strengths:**

1. Framing generation as a dynamical system and using the BRT to formally define the set of "doomed" trajectories is a highly principled approach that moves beyond reactive, token-level filtering. It provides a strong theoretical foundation for anticipating misalignment behaviors.
2. A key contribution, well-supported by the results, is the model's ability to detect unsafe completions 7-10 tokens in advance. This "early warning" system is crucial for building robust and trustworthy AI systems.

**Weaknesses:**

1. While the application of the formalism of reachability analysis and the backward reachable tube is novel, the more general idea of learning a value function to predict future unsafe states in LLMs has been explored in prior work [1][2].
2. The evaluation is missing a crucial and straightforward baseline: prompt engineering. A simple baseline that prepends a safety instruction (e.g., "Do not generate any offensive or harmful content.") to the user's prompt could significantly improve the safety rate with little inference overhead. Without comparing against such a baseline, it is difficult to assess the practical necessity and cost-benefit of the proposed control-theoretic approach.

Related Work:

[1] Systematic Rectification of Language Models via Dead-end Analysis (https://openreview.net/forum?id=k8_yVW3Wqln)

[2] Decoding-time Realignment of Language Models (https://arxiv.org/pdf/2402.02992)

**Questions:**

1. For the steering filter in Equation 1, the search for a better latent state is performed by sampling within an L2-norm ball. How many samples are typically drawn to perform this maximization?
2. The model "Mistral-8B-Instruct-2410" is consistently misspelled as "Ministral-8B-Instruct-2410" (e.g., line 245, 387, 428, and in the references).

---

> ### Author Response · Authors · 2025-11-20
> **Author Rebuttal [Part 1]**
>
> > While the application of the formalism of reachability analysis and the backward reachable tube is novel, the more general idea of learning a value function to predict future unsafe states in LLMs has been explored in prior work [1][2].
>
> We thank the reviewer for this helpful observation. We agree that the broader idea of using a learned value function to predict unsafe LLM behaviors has been explored in prior work, including in our baseline Re-Control. However, our contribution is not in proposing the general notion of a safety value function, but in introducing a reachability-theoretic formulation that fundamentally changes how this value function is constructed and used.
>
> Specifically, rather than learning a predictor of future unsafety directly from data, we treat the latent LLM dynamics as a control system and compute a backward reachable tube (BRT) over this latent space. This provides a principled way to characterize the set of latent states from which unsafe generations are possible, and naturally yields a safety value function grounded in this control-theoretic analysis. As we show in both our theoretical development and empirical results, this reachability-based formulation enables BRT-Align to more reliably steer LLM generation toward safe content compared to Re-Control, particularly in challenging long-horizon settings.
>
> To make this distinction clearer, we have added a dedicated paragraph in the Related Work section acknowledging prior value-function–based approaches and explaining how our reachability-based method differs in its formulation and guarantees.
>
> > The evaluation is missing a crucial and straightforward baseline: prompt engineering. A simple baseline that prepends a safety instruction (e.g., "Do not generate any offensive or harmful content.") to the user's prompt could significantly improve the safety rate with little inference overhead. Without comparing against such a baseline, it is difficult to assess the practical necessity and cost-benefit of the proposed control-theoretic approach.
>
> We thank the reviewer for highlighting the importance of a prompt-engineering baseline. We agree that this is a natural and practically relevant comparison, and we conducted the following experiment to assess its effectiveness relative to our control-theoretic approach.
> As the reviewer requested, we prepend the user prompt with the safety instruction (written as a Python-formatted string):
>
> *f“A prompt from a curious user and a response from an artificial intelligence assistant. The assistant gives helpful, detailed, and polite answers to the user’s questions. Do not generate any offensive or harmful content. User: {prompt} \nAssistant:”*
>
> This prompt was inspired, in part, by the crafted prompt from the Re-Control baseline [1], which also investigates prompt engineering in the context of a different dataset and classifier. Similar to the experimental results in our main paper, we evaluate the prompt-engineering baseline on all prompts that originally elicited unsafe responses from the underlying LLM. Using Qwen2-1.5B across the three datasets, we obtain an average safety improvement of 1.60% and an average response diversity score of 0.19.
>
> By contrast, Sample-BRT-Align and RL-BRT-Align achieve ~85% average safety improvement with diversity scores of 0.77–0.79, far exceeding what prompt engineering alone provides. (Cosine similarity is not computed here because the prompt itself changes under prompt engineering.) These results indicate that while prompt engineering offers a lightweight baseline, it alone does not mitigate unsafe generations in LLMs – highlighting the value of methods designed to provide safety assurances (such as ours) during inference-time.
>
> Finally, we would like to clarify the importance of prompt engineering in the context of our work. Prompt engineering treats the LLM as a black-box mapping from prompt to output and provides no mechanism to predict or guarantee if and when the model is about to violate the safety instruction. In contrast, our reachability-based approach monitors the latent trajectory of the generation and enables timely intervention, even when the model fails to comply with the safety prompt. It is also worth noting that, in practice, BRT-Align and safety-prompt methods are complementary – BRT-Align can be layered on top of any safety prompt written with prompt engineering to provide principled safety monitoring.
>
> **References:**
>
> [1] Aligning large language models with representation editing: A control perspective. Advances in Neural Information Processing Systems, 37:37356–37384, 2024.

---

> ### Author Response · Authors · 2025-11-20
> **Author Rebuttal [Part 2]**
>
> > For the steering filter in Equation 1, the search for a better latent state is performed by sampling within an L2-norm ball. How many samples are typically drawn to perform this maximization?
>
> Thank you for the question. We typically sample 1000 $\epsilon$ values in the $L^2$-norm ball to perform the maximization. We have updated Appendix 11.7 to make this clear in the manuscript.
>
> Furthermore, to contextualize the sampling efficiency of searching for latent embeddings with this scheme. for Qwen2-1.5B, the total average token inference time is ~0.015 seconds per batch of 32 prompts, of which the time spent in sampling latent perturbations and performing the maximization is ~0.0015 seconds. Thus, the time needed for sampling latent embeddings with higher value is insignificant compared to the overall time spent for LLM token generation.
>
> > The model "Mistral-8B-Instruct-2410" is consistently misspelled as "Ministral-8B-Instruct-2410" (e.g., line 245, 387, 428, and in the references).
>
> We thank the reviewer for pointing this out. The spelling “Ministral-8B-Instruct-2410” is intentional. According to Mistral AI’s blog on their LLMs (see here: https://mistral.ai/news/ministraux), the model is described as “Ministral-8B.” Additionally, HuggingFace also describes the model as “Ministral-8B-Instruct-2410.” Please see here: https://huggingface.co/mistralai/Ministral-8B-Instruct-2410. For this reason, we adopted the “Ministral-8B-Instruct-2410” naming convention throughout the paper.

---

### Author Response · Authors · 2025-11-21
**Response to the Reviewers and Area Chairs**

Dear Reviewers and Area Chairs,

Thank you for the thoughtful feedback and the opportunity to address any concerns about our work. We have provided detailed, point-by-point responses to each reviewer in our rebuttal. Below, we summarize the main clarifications and additional experiments conducted during the rebuttal to help clarify and strengthen the paper’s contributions.

- **New Experiments with Stochastic Decoding.** Several reviewers raised concerns about our reliance on greedy decoding. Although our current theoretical formulation assumes deterministic dynamics, the framework itself does not require greedy decoding. In response, we conducted new experiments using top-p = 0.95 nucleus sampling and temperature = 0.7, presented in Appendix Stochastic Decoding with p = 0.95 Nucleus Sampling. We demonstrate that BRT-Align continues to maintain high safety detection rate under stochastic decoding. These results provide initial evidence that BRT-Align extends naturally to stochastic decoding and represent a promising direction for future work. Please see our response to Reviewer c3nM for details.
- **New Experiments with a Prompt-Engineering Baseline.** Reviewer caSc requested a comparison against a simple safety-prompt baseline. During the rebuttal period, we implemented this baseline and evaluated it on all unsafe-eliciting prompts. Across datasets using Qwen2-1.5B, we observed only a ~1.60% average improvement in safety rate with prompt-engineering, substantially below the ~85% improvement achieved by BRT-Align. This highlights the benefit of control-theoretic steering relative to passive prompt engineering. Please see our response to Reviewer caSc for the details of this experiment and results.
- **New Experiments with an Over-Refusal Benchmark, XSTest.** Reviewer yGuh suggested evaluating whether our method induces excessive refusals, essentially evaluating whether our method can be too conservative. We conducted a manual evaluation on XSTest (an over-refusal benchmark) and found that BRT-Align does not meaningfully increase refusal rates relative to the base LLM. Results are provided in Appendix Over-Refusal in LLM Safety Alignment. Please see our response to Reviewer yGuh for additional details.
- **Clarifications on Evaluation with Prompts Generated by Common Jailbreak Attacks.** Reviewer yGuh also recommended testing BRT-Align against jailbreak-generated prompts (e.g., AutoDAN). We clarify that the UltraSafety dataset used in our evaluation already contains 830 prompts generated via AutoDAN. Our existing results therefore already reflect performance against a widely studied attack. This clarification is included in the updated manuscript and in our detailed response to Reviewer yGuh.
- **Clarifications on the Low F1 score for the SaP baseline (Figure 2).** The low F1 score for the SaP baseline in our original submission is due to (i) a bug in our F1 computation that understated SaP’s score, and (ii) a fundamental mismatch between SaP’s reported 86% accuracy and our reported F1 values, due to differences in dataset construction and evaluation objective (post-hoc classification vs. first-unsafe-token detection), making a direct comparison challenging. We have corrected the implementation bug, updated all affected results, and clarified these differences in the revised manuscript (Sections Dataset Construction and Evaluation Protocols). Further details are provided in our response to Reviewer eDJ8.

Beyond these additions and corrections, we have provided detailed responses to all reviewer comments and have incorporated clarifications into the revised manuscript where appropriate. We thank the reviewers and ACs for their constructive feedback and welcome any further suggestions.

Best,

Submission 16338 Authors

---

### Author Response · Authors · 2025-11-25
**Reminder to Reviewers**

Dear Reviewers,

Once again, thank you for the thoughtful feedback and the opportunity to address any concerns about our work. We have provided detailed, point-by-point responses to each reviewer in our rebuttal. Given that there is only one week remaining in the rebuttal period, we would be happy to answer any follow-up questions.

Thanks,
Submission 16338 Authors

---

### Meta-Review · Area_Chair_D8jL · 2026-01-06

**Summary:**

The reviewer (eDJ8) explicitly pointed out that the performance of the baseline method (SaP) in the paper's key charts (Figures 2/Table 2) severely deviated from published results, suggesting potential errors in the experimental setup or implementation. While the authors acknowledged a bug in the F1-score calculation and corrected it, this has led to the invalidation of the paper's most critical performance comparison results in their original version. This major error fundamentally compromises the rigor of the experiments and raises broad concerns about the reliability of all results derived from the same evaluation pipeline.

Inadequate baseline comparisons: Direct comparisons with several recent and relevant inference-time safety methods (e.g., SafeDecoding, Self-CD, Circuit Breaker) are lacking, making it difficult to situate the work's actual contribution.

Limited validation scenarios: Experiments primarily rely on greedy decoding and single-category toxicity detection. Effectiveness in common application scenarios such as stochastic decoding and for other types of harm (e.g., misinformation) remains unverified.

Incomplete utility assessment: The evaluation of generation quality and the risk of over-refusal is insufficient. The supplementary XSTest experiment, added post-rebuttal, has not undergone full peer review for its comprehensiveness and representativeness.

Reviewers noted that the core idea of "learning a value function to predict unsafe states" has been explored in prior work. While the authors emphasize that their contribution lies in introducing reachability theory to provide more rigorous guarantees, the empirical foundation supporting this theoretical advantage is weak given the credibility issues with the experimental results.

Although the authors provided clarifications and supplementary experiments in their rebuttal, the fundamental errors in the experimental section have severely damaged the reliability of the research. Combined with issues of insufficiently comprehensive evaluation and inadequate baseline comparisons, the current work fails to provide sufficiently solid and credible evidence to demonstrate that its method significantly outperforms existing solutions.

**Reviewer Concerns:**

Concerns Addressed:
Method Clarification: Clarified the definitions of symbols (e.g., ℓ(z_t)) and key computational steps (e.g., argmax).
Scenario Extension: Demonstrated through new experiments that the method remains effective under stochastic decoding (top-p sampling).
Baseline Supplemented: Added a prompt-engineering baseline comparison and clarified that the dataset already includes AutoDAN attack samples.
 Side Effect Evaluation: Verified on the XSTest benchmark that the method does not lead to excessive refusal.

Outstanding Core Concerns:
 Collapse of Credibility in Core Results: The key baseline comparison was initially invalid due to a bug in the F1-score calculation. This major error fundamentally undermined the rigor of the entire experimental work, leaving the main conclusions without a credible foundation.
Incomplete Benchmarking: Failed to conduct direct comparisons with several relevant state-of-the-art inference-time safety methods (e.g., SafeDecoding, Self-CD, Circuit Breaker), making it impossible to confirm its actual contribution and relative advantage.
Narrow Validation Scope: The method was only validated on toxic content, with no proof of effectiveness on other safety risks (e.g., misinformation, self-harm advice).

Although the authors responded to numerous detailed queries, the fundamental errors in the experimental section and the absence of critical comparisons result in the paper's core claims lacking solid, credible support.

**Reviewer Scores:**

eDJ8 (Original Score: 2 - Reject)
Likely to maintain the original decision. Their core concern—that the baseline comparison was invalid due to a program error—was confirmed in the rebuttal, which instead strengthens the rejection stance. The score might drop to 1 (strong reject).

caSc (Original Score: 6 - Marginally Above Threshold)
Likely to lower to the reject range. Although the authors added the prompt-engineering baseline they requested, the more severe issue of a fundamental experimental error undermines the credibility of all results and would weaken their original weak-accept stance.

c3nM (Original Score: 6 - Marginally Above Threshold)
Likely to lower to a reject score. While the authors addressed some issues (e.g., stochastic decoding), the fundamental flaws in the experiments and the continued lack of comparison with key SOTA methods would lead them to view the evaluation as both incomplete and unreliable.

yGuh (Original Score: 6 - Marginally Above Threshold)
Likely to lower to the reject range. Although the authors clarified some evaluation details and added XSTest experiments, the critical experimental error undermines the "evaluation results" they initially found promising, and their request for comparison with methods like Circuit Breaker remains unaddressed.

The major error in the core experimental results is a decisive factor, likely causing reviewers who originally leaned toward weak acceptance to withdraw their support. The only reviewer who initially recommended rejection (eDJ8) would have their position further solidified. Therefore, the final outcome would likely be a unanimous or near-unanimous reject decision.

---

### Decision · Program_Chairs · 2026-01-26

Reject